# Genetic Transformation and Green Fluorescent Protein Labeling in *Ceratocystis paradoxa* from Coconut

**DOI:** 10.3390/ijms20102387

**Published:** 2019-05-14

**Authors:** Xiaoqing Niu, Mengtian Pei, Chenyu Liang, Yuexiao Lv, Xinyi Wu, Ruina Zhang, Guodong Lu, Fengyu Yu, Hui Zhu, Weiquan Qin

**Affiliations:** 1Key Laboratory of Biopesticide and Chemical Biology, Ministry of Education, Fujian Agriculture and Forestry University, Fuzhou 350002, China; xiaoqingniu123@126.com (X.N.); 15870651077@163.com (M.P.); liangcy@163.com (C.L.); lyuyuexiao513@163.com (Y.L.); Yimi1207@163.com (X.W.); 18838933694@126.com (R.Z.); 2Coconut Research Institute, Chinese Academy of Tropical Agricultural Sciences, Wenchang 571339, China; yufengyu17@163.com (F.Y.); zhuhui@catas.cn (H.Z.); qwq268@163.com (W.Q.)

**Keywords:** stem-bleeding disease of coconut, *Ceratocystis paradoxa*, *gfp* gene, transformation, protoplast

## Abstract

*Ceratocystis paradoxa*, the causal agent of stem-bleeding disease of the coconut palm, causes great losses to the global coconut industry. As the mechanism of pathogenicity of *C. paradoxa* has not been determined, an exogenous gene marker was introduced into the fungus. In this study, pCT74-sGFP, which contains the green fluorescent protein (GFP) gene, and the hygromycin B resistance gene as a selective marker, was used as an expression vector. Several protoplast release buffers were compared to optimize protoplast preparation. The plasmid pCT74-sGFP was successfully transformed into the genome of *C. paradoxa*, which was verified using polymerase chain reaction and green fluorescence detection. The transformants did not exhibit any obvious differences from the wild-type isolates in terms of growth and morphological characteristics. Pathogenicity tests showed that the transformation process did not alter the virulence of the X-3314 *C. paradoxa* strain. This is the first report on the polyethylene glycol-mediated transformation of *C. paradoxa* carrying a ‘reporter’ gene GFP that was stably and efficiently expressed in the transformants. These findings provide a basis for future functional genomics studies of *C. paradoxa* and offer a novel opportunity to track the infection process of *C. paradoxa*.

## 1. Introduction

Coconut palm (*Cocos nucifera* L.) is one of the most economically important trees in the world. It generates employment and income in many countries, where its fruit is either eaten raw or processed into manufactured products and by-products. However, coconut production is affected by many biotic and abiotic factors. Coconut stem-bleeding (CSB), caused by the fungus *Ceratocystis paradoxa* (De Seynes) Höhn (anamorph *Thielaviopsis paradoxa*), is one of the most notorious biotic diseases of coconut. CSB is known to occur in coconut-producing areas worldwide. The disease was first reported in Sri Lanka [1] and caused severe damage to PB-121 hybrids in Indonesia. In the state of Sergipe, Brazil, it was first detected in early 2004 [2]. It was discovered in China for the first time in 2009, where it was found to occur in nearly all coconut gardens [3]. The disease has gradually become the primary concern of producers and researchers due to its rapid spread and great lethality to coconut. Affected plants perish within 3–4 months after the first appearance of stem symptoms.

The symptoms of CSB are described as follows: The affected trunk areas exhibit dark discoloration and a reddish-brown or rust-colored liquid bleeding from the stem cracks that may turn blackish when dried, resulting in rotting when moist; there is a reduced frequency of leaf emergence and a reduced young leaf size, with stem-thinning occurring near the canopy as the disease progresses; and the leaves eventually become brownish-yellow and fragile [2,3]. Besides the trunk and leaves, *C. paradoxa* also infects the coconut fruit, with the infected pericarp becoming black and soft and producing a strong fruity aroma [4].

The ascomycete *C. paradoxa* not only infects palm trees, such as *Cocos nucifera*, *Phoenix dactylifera*, *Borassus flabellifer*, *Butia capitata*, *Elaeis guineensis*, and *Areca catechu*, but also infects other plants, including *Ananas comosus*, *Mangifera indica*, *Musa paradisiaca*, *Solanum muricatum*, *Theobroma cacao*, and *Zea mays*, resulting in large agricultural losses [3]. The biological characteristics of the pathogen have thus far been described [5], and the partial and temporal dynamics of stem-bleeding have been recorded for clarifying the disease progression and characterizing the production losses from the disease [6]. In addition, some CSB disease management studies have been reported, including physical, chemical, and biological control measures [7,8,9,10]. However, a detailed understanding of the pathogenesis and infection dynamics of this pathogen is lacking. While a previous report on the transformation of *C. paradoxa* using *Agrobacterium* exists [11], the pathogenesis of *C. paradoxa* was not explored.

Highly efficient transformation would be beneficial for a detailed investigation of the infection characteristics and pathogenesis of *C. paradoxa* and its interaction with the host. Green fluorescent protein (GFP) expression has proven to be a useful tool for such analyses in filamentous fungi [12,13] and has been successfully expressed in numerous ascomycetes, including *Colletotrichum acutatum*, *Verticillium fungicola Acremonium chrysosporium*, and *Sordari macrospora* [14,15]. In this study, the first protoplast preparation and polyethylene glycol (PEG)-mediated transformation system for *C. paradoxa* was established, and the expression vector pCT74-sGFP was transformed into the fungus to obtain a recombinant fungus labeled with GFP. Our protocol will inform research that involves monitoring the infection progression of this fungus in the coconut and should also inform the further exploration of the mechanisms of *C. paradoxa* pathogenicity.

## 2. Results

### 2.1. Sensitivity of C. paradoxa to HmB

Hyphal inocula of the wild-type strain X-3314 were inoculated on potato dextrose agar (PDA) plates containing different concentrations of hygromycin B (HmB). After five days, the fungus continued to grow on the plates when the concentration of HmB remained below 30 μg/mL, but failed to grow at concentrations higher than 40 μg/mL (Figure 1). Thus, 30 μg/mL HmB could markedly inhibit the growth of *C. paradoxa*, while 40 μg/mL HmB completely inhibited growth. PDA containing 40 μg/mL HmB was therefore used to screen for successful transformants carrying a functional *hph* gene. In this study, each plate consisted of 10 mL PDA with 2 μL, 4 μL,6 μL,8 μL of HmB (1 g/20 mL), resulting in final concentrations of HmB of 10 μg/mL, 20 μg/mL, 30 μg/mL, and 40 μg/mL.

### 2.2. Protoplast Preparation

Following digestion with protoplast release buffer 2 at 31 °C for 30 min, the mycelia began to rupture and single cells could be observed. Sufficient protoplasts began to form after an additional 1 h of digestion (Figure 2), with the concentration reaching 3.0 × 10^9^ spores mL^−1^. Compared with this protoplast release buffer, the quantity of protoplasts obtained using the other protoplast release buffers (buffers 1, 3, 4, 5, and 6) was far below 3.0 × 10^9^. Furthermore, the other buffers proved time-consuming and wasteful.

### 2.3. Screening and Detection of Transformants

The *C. paradoxa* transformants were selected using 100 mL of selective medium containing 80 μL of 1 g/20 mL HmB. Five days after transformation, the transformants were inoculated on fresh PDA plates containing 160 μL of 1 g/20 mL HmB. Together with the results of several transformation cases, the average transformation efficiency was 4~5 transformants per μg DNA. Four transformants (X1, X2, X3, and X4) were selected for further study. These four transformants could grow normally on CM plates containing 16 μL of 1 g/20 mL HmB. Whereas the wild-type failed to grow (Figure 3), the plates contained 10 mL CM, which indicated that the four transformants possessed the hygromycin B resistance gene. The colonies and hyphal morphology of the four transformants were almost identical to the wild-type. After subculturing seven times on PDA plates lacking HmB, the four transformants were still able to grow on the HmB PDA plate. This suggested that all four transformants exhibited genetic stability.

The X1, X2, X3, and X4 transformants showed strong fluorescence when observed under the confocal microscope (Figure 4), and their fluorescence remained genetically stable after seven subcultures on PDA medium. This result suggested that the GFP gene could be stably expressed in *C. paradoxa*.

### 2.4. PCR Confirmation of the Transformants

To verify that the GFP gene had been integrated into the genome, the genomic DNA of X1, X2, X3, and X4 was amplified using the ToxAsGFPF/ToxAsGFPR primers. The predicted 1130 bp fragment was amplified from the transformants and plasmid control, but never from the wild-type strain. This indicated that the GFP gene had been integrated into the *C. paradoxa* transformants.

### 2.5. Pathogenicity Detection of the Transformants

The GFP-labeled transformants were used to infect wounded coconut fruits. Three days after inoculation, characteristic disease symptoms appeared on the fruits that had been inoculated with the wild-type strain X-3314, as well as the X1, X2, X3, and X4 transformants (Figure 5). The *C. paradoxa* X-3314 strain and the transformants were re-isolated successfully from the diseased tissues. Additionally, the isolated transformants still exhibited strong green fluorescence (Figure 6). These findings suggest that the transformants can be used to study the infection process of this fungus.

## 3. Discussion

As the causal agent of coconut stem-bleeding (CSB), *C. paradoxa* causes great economic losses to the coconut industry. Understanding the invasion, colonization, and localization mechanisms of *C. paradoxa* in coconut could facilitate the development of new strategies for disease control in the future. In the present study, we successfully transformed the hygromycin B resistance gene and a ‘reporter’ *gfp* gene into *C. paradoxa*, obtaining stable transformants of *C. paradoxa* that efficiently expressed GFP.

Protoplast quantity is one of the key factors influencing transformation efficiency. The results of this study demonstrated that transformants were rarely obtained when the number of protoplasts was lower than 1 × 10^6^/mL^−1^. This corroborates the consensus that a high concentration of protoplasts is required for successful transformation [16,17]. The concentration of the plasmid pCT74-sGFP also influenced the transformation results, with 500 ng/μL of pCT74-sGFP means approximately 2 μg of DNA being required. Thus, the preparation of a sufficient quantity of protoplasts is necessary for successful transformation, but a high plasmid concentration is equally important.

Due to fungal cell wall differences, the enzyme type and digestion conditions can affect protoplast production. In this study, the cell wall of *C. paradoxa* was digested with different protoplast release buffers at the same temperature (31 °C), but the duration of the treatment and the quantity of the protoplasts obtained differed between the buffers. A comparison of the six treatment results indicated that adequate protoplasts could only be obtained after digestion with protoplast release buffer 2 for 90 min, while the other treatments (protoplast release buffer 1, 3, 4, 5, and 6) were either overly time-consuming or produced lower yields.

After subculturing seven times on PDA plates lacking HmB, the four transformants (X1, X2, X3, and X4) could still grow on HmB PDA plates and exhibited high green fluorescence inside the hyphae and conidia, which is indicative of genetic stability. The results indicated that the *gfp* gene had been successfully transformed into *C. paradoxa*. Furthermore, the pathogenicity test demonstrated that the transformants caused typical disease symptoms that were the same as the wild-type strain, which suggested that the pathogenicity of the transformant strain had not changed. This verified that they can be used for studying *C. paradoxa* infection in coconut palms.

GFP as a reporter has been widely used in fungi [18,19,20]. The advantage of real-time observation renders it a great tool for the analysis of living cells and organisms. However, its application is limited in plant pathogen analyses due to the lower number of transformed species or inefficient expression and low fluorescence [21]. The successful expression of GFP is usually based on a stronger promoter or a high copy number viral vector [22]. The ToxA promoter has proven valuable for the strong expression of GFP in ascomycetes with diverse lifestyles [23]. Our results indicated that this promoter can be used to obtain high GFP expression in *C. paradoxa*.

This study is the first to report on GFP expression in *C. paradoxa*. The transformants can be used for the study of fungal colonization, localization, and invasion in plants. Furthermore, the protoplast preparation and transformation method described here can be used as a tool for efficiently constructing a random insertional transformant library, gene functional analysis, and the exploration of the pathogenic mechanisms of *C. paradoxa*.

## 4. Materials and Methods

### 4.1. Strain and Plant Material

The *C. paradoxa* X-3314 strain (Collection No. CCTCC AF2014002) was isolated from an infected coconut stem and stored at the Coconut Research Institute of Chinese Academy of Tropical Agricultural Sciences (Wenchang, Hainan province, China). The *Escherichia coli* strain DH5a was used in this study. The plant material was coconut fruits of “Malayan Red Dwarf”.

### 4.2. Fungal Transformation Vector

A pCT74-sGFP plasmid (Figure 7) was constructed and stored at the Functional Genomics Center, Fujian Agriculture and Forestry University. The transformation construct relies on antibiotic resistance conferred by a modified form of the *E. coli* hygromycin B phosphotransferase (*hph*) gene for transformation selection [24]. PCT74 further employs the necrosis-inducing host-selective toxin gene ToxA promoter from *Pyrenophoratritici repentis* to drive the expression of the synthetic GFP genes (sGFP) [22,25].

### 4.3. Media and Reagents

The media used in this test included complete liquid medium (CM) (6 g yeast extract, 10 g sucrose, 6 g casein acid hydrolysate, 15 g agar, 1000 mL ddH_2_O), potato dextrose agar (PDA) solid medium (200 g potato, 20 g sucrose, 15 g agar, 1000 mL ddH_2_O), starch yeast liquid medium (SYM) (2 g yeast extract, 10 g starch, 3 g sucrose, 1000 mL ddH_2_O), Luria-Bertani (LB) solid medium (5 g yeast extract, 10 g peptone, 10 g NaCl, 15 g agar, 100 mg/mL ampicillin Na (Amp, Takara), 1000 mL ddH_2_O), TB3 liquid medium (3 g yeast extract, 3 g casamino acid, 20% sucrose, 1000 mL ddH_2_O), TB3 solid medium (3 g yeast extract, 3 g casamino acid, 20% sucrose, 15 g agar, 1000 mL ddH_2_O).

STC buffer (44 g sorbitol, 10 mL l.0 mol/L Tris-HCL (pH 8.0), 10 mL 1 mol/L CaCl_2_, 200 μL streptomycin, add ddH_2_O up to 200 mL, and sterile water filtered through a 0.22-μm filter (Jet Biofil FPV203000)). PTC buffer (4 g PEG3350, 10 mL STC, 10 μL streptomycin nitrate, heat-dissolved at 60°C for 20 s and sterile filtered), osmotic pressure stabilizer (1.0 mol/L mannitol, 1.0 mol/L sorbitol), protoplast release buffer 1 (1% lysing enzyme (Sigma-Aldrich, St. Louis, MO, US, CB8709259) with 1.0 mol/L mannitol); protoplast release buffer 2 (1% Driselase (Sigma-Aldrich, cat. No. CB0296810) with 1.0 mol/L mannitol in STC); protoplast release buffer 3 (mixture of 1% Driselase and 1% lysing enzyme with 1.0 mol/L mannitol in STC); protoplast release buffer 4 (1% lysing enzyme with 1.0 mol/L sorbitol in STC); protoplast release buffer 5 (1% Driselase with 1.0 mol/L sorbitol in STC); protoplast release buffer 6 (mixture of 1% Driselase and 1% lysing enzyme with 1.0 mol/L sorbitol in STC) were used to prepare protoplasts from mycelia. The hygromycin B (HmB) was purchased from Takara Co. The PCR primers used in this test were synthesized by Sangon Biotech (Shanghai), Co., Ltd., Shanghai, China.

### 4.4. Sensitivity of C. paradoxa to HmB

Hygromycin B (HmB) ranging from 10–40 μg/mL was used for testing the sensitivity of *C. paradoxa* to the antibiotic. Agar blocks (5 mm × 5 mm) containing mycelia were cut from the edge of the *C. paradoxa* colonies grown on PDA plates and transferred into the center of new PDA plates supplemented with different concentrations of hygromycin B and incubated at 25 °C, PDA plates without HmB served as a control. The effect of HmB on *C. paradoxa* growth was analyzed by measuring the diameter of growing mycelial on the plates after 5 days. The test was carried out in triplicate.

### 4.5. Plasmid Extraction of pCT74-sGFP

The *E. coli* DH5α construct containing pCT74 was spread onto LB plates containing ampicillin Na (Amp, 100 μg/mL). A single colony was inoculated into LB liquid medium containing 2 μL 100 mg/mL Amp and incubated with shaking at 37 °C overnight. Plasmid extraction was achieved using a plasmid Mini Kit (Takara, Japan). Plasmids were detected by 1.5% agarose gel electrophoresis and the plasmid DNA concentration was measured using a Micro-Ultraviolet Spectrophotometer (NanoDrop 2000, Thermo Scientific, Waltham, MA, USA).

### 4.6. Protoplast Preparation

The entire procedure detailed below was performed under aseptic conditions. A 5-mm diameter agar disc from the growth margins of the X-3314 colony was placed in the center of a fresh PDA plate. After incubation at 25 °C for 5 days, the colony had grown over and covered the plate. Ten milliliters of sterile water was added to the plate and the conidia of *C. paradoxa* were collected and adjusted to a concentration of 1 × 10^7^ spores mL^−1^ with sterilized water using a hemocytometer prior to inoculation. Approximately 6 mL of l × 10^6^ spores mL^−1^ conidia was inoculated into the SYM medium and distributed into three flasks, with 400 mL of medium per flask. The flasks were incubated at 25 °C for 24 h under shaking at 60 rpm/min. The mycelia were collected using two layers of micro-cloth (Calbiochem, Cambridge, MA, USA) and washed with sterilized water twice, followed by washing three times with 15 mL osmotic pressure stabilizer, and then dried on filter paper. One gram of mycelia was collected and thoroughly suspended in protoplast release buffer 2 at 31 °C with shaking at 90 rpm/min for 90 min. Three layers of micro-cloth were placed in a 50-mL sterilized centrifuge tube used for filtering the lysis solution, after which the filtrates were centrifuged at 4000 rpm for 10 min at 4 °C. After discarding the supernatant, the pellet was washed by gently re-suspending it in 20 mL STC and the same STC wash procedure was repeated twice. The concentration of the protoplasts was then adjusted to 6.0 × 10^7^–3.0 × 10^8^/mL with STC. Finally, 20 μL DMSO was added to the protoplast suspension, which was then distributed into 2 mL centrifuge tubes with 150 μL of suspension per tube. The tubes were then stored at −80 °C until use.

### 4.7. Protoplast Transformation

Ten microliters of pCT74-sGFP plasmid DNA was added to 150 μL of prepared protoplast suspension and gently mixed, and then incubated at 25 °C for 30 min. Following this, 600 μL PTC was added and the solution was incubated at room temperature (25 °C ± 1 °C) for 30 min, after which 6 mL TB3 was added to the reaction tube (50 mL) and incubated at 25 °C with shaking (90 rpm) overnight. The treatment tube with liquid TB3 was mixed with 100 mL solid TB3 medium (that had been melted at 50 °C) containing 80 μL of a 40 μg/mL HmB stock solution and poured into 15 cm diameter plates, which were then incubated at 25 °C for 3–4 d in preparation for transformant growth detection.

### 4.8. Screening for HmB Resistance and Stability Test

The potential transformant colonies growing on HmB plates were used to inoculate selective PDA medium containing 80 μL of a 40 μg/mL HmB stock solution, with one plate for each potential transformant. These plates were then incubated at 25 °C for 4~5 days. To confirm the stability of the transformants, they were inoculated on PDA plates lacking HmB and incubated at 25 °C for 3 d. The mycelia from the edge of the colonies were inoculated on fresh PDA plates for seven successive subcultures, and then transferred to selective PDA plates containing 80 μL of a 40 μg/mL HmB stock solution in order to observe the growth under selective conditions. The fluorescence of the transformants was monitored using a confocal microscope (Nikon C-HGFIE) with VC40 oil objective, and fluorescence was detected at 488 nm.

### 4.9. PCR Detection

The primers were as follows: ToxAsGFPF, 5’TGGAATCCATGGAGGAGTTC3’; ToxAsGFPR, 5’CTTGTACAGCTCGTCCATGC3’. To determine whether the *gfp* gene had integrated into the genome, a pair of primers (ToxAsGFPF and ToxAsGFPR’) were designed based on the ToxA promoter sequence and the sGFP sequence. Polymerase Chain Reaction (PCR) was used to confirm the integration of the GFP gene insert, with a predicted length of the amplified sequence containing the GFP of about 1.1 kb. The genomic DNA of the transformants and wild-type strain was extracted using a modified cetyltrimethylammonium bromide (CTAB) method [26]. The pCT74-sGFP plasmid was regarded as a positive control. The PCR reaction system contained: 2.5 μL 10×PCR buffer, 2.5 μL 2.5 mmol/L dNTPs, 1.0 μL primerF, 1.0 μL primerR, and 0.5 μLExtag. The PCR reaction procedure was as follows: 95°C for 3 min, followed by 35 cycles of 94 °C for 40 s, 64 °C for 40 s, 72 °C for 80 s, and a final extension at 72 °C for 10 min.

### 4.10. Pathogenicity Detection of the Transformants

The wild strain X-3314 and positive transformants were cultured on PDA medium at 25 °C for 5 d. Coconut fruits (“Malayan Red Dwarf”) were wounded and inoculated with a 1.5 × 10^5^ spores·mL^−1^ suspension of X-3314 or transformants. Wound incisions were made with a sterilized cork borer. The experiment was repeated in triplicate.

## Figures and Tables

**Figure 1 ijms-20-02387-f001:**
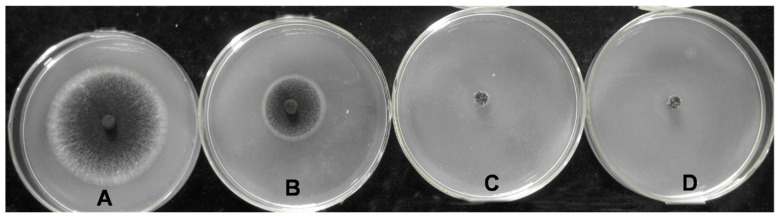
The growth of *C. paradoxa* on PDA medium with HmB 10 ug/mL, (**A**); 20 ug/mL, (**B**); 30 μg/mL, (**C**); 40 μg/mL, (**D**).

**Figure 2 ijms-20-02387-f002:**
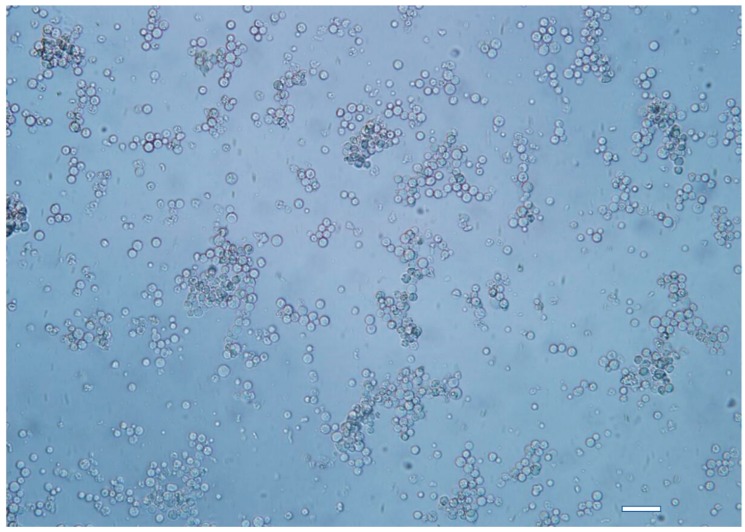
The mycelia were digested in protoplast release buffer 2 at 31 °C with shaking for 1.5 h. Sufficient protoplasts formed thereafter, with concentrations reaching 3.0 × 10^9^ spores mL^−1^. Bar: 100 μm.

**Figure 3 ijms-20-02387-f003:**
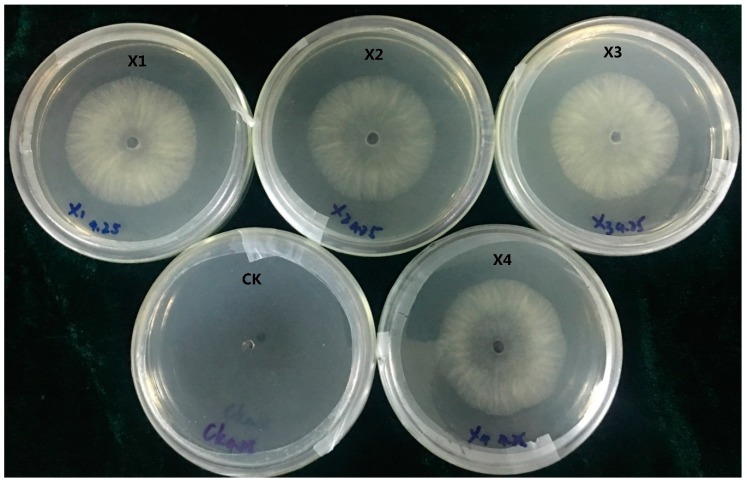
The growth of the transformants on HmB plates. (**CK**), wild-type strain X-3314; (**X1**–**X4**), transformants. The transformants and wild-type strain were cultured on PDA plates containing 16 μL of 1 g/20 mL HmB, respectively. The figure shows that the transformant strains (**X1**–**X4**) could grow on PDA plates, while the wild-type failed to grow.

**Figure 4 ijms-20-02387-f004:**
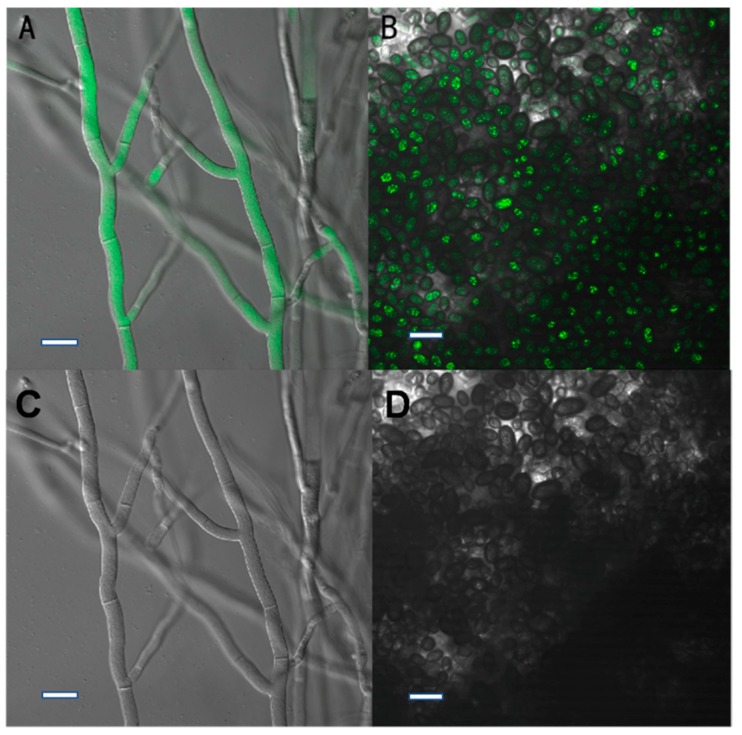
Expression of GFP in the transformants. (**A**,**B**), green fluorescence was observed in the mycelia and conidia of transformant X1 using a confocal microscope; (**C**,**D**), the mycelia and conidia in visible light. Bars: **A**–**D** = 10 μm.

**Figure 5 ijms-20-02387-f005:**
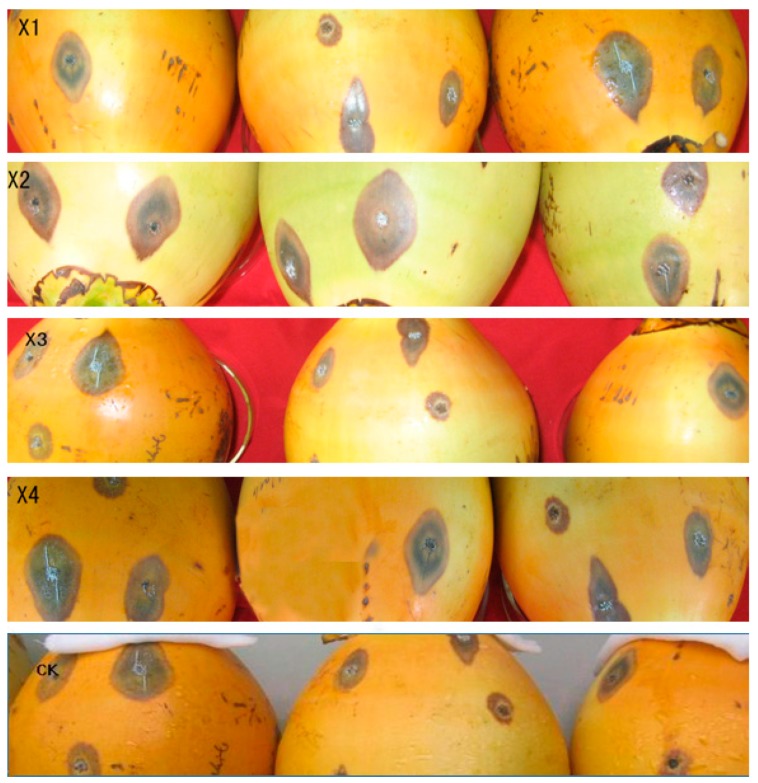
Pathogenicity detection of the transformants. Symptoms of coconut stem-bleeding disease on coconut fruits after wound-inoculating 1.5 × 10^5^ spores·mL^−1^ suspension of transformants (**X1**–**X4**), respectively, for 3 days. CK: wild-type strain X-3314 was used as a control.

**Figure 6 ijms-20-02387-f006:**
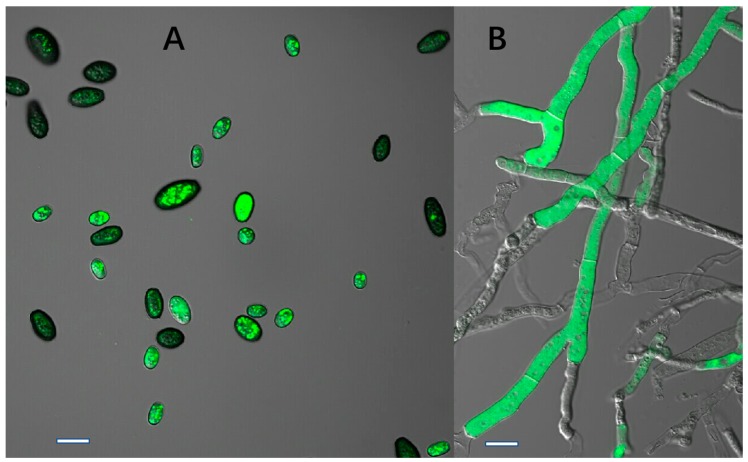
The isolated transformants exhibited strong green fluorescence in hypha and conidia. Bars: (**A,B**) = 10 μm.

**Figure 7 ijms-20-02387-f007:**
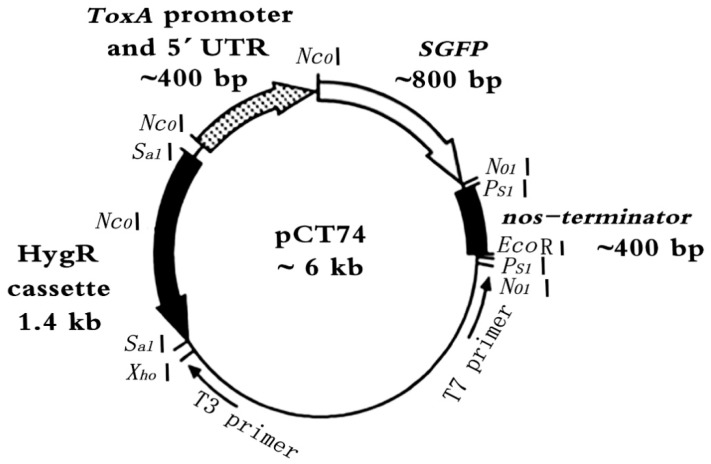
Map of expression vector for filamentous fungi pCT74-sGFP.

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
