# Peer review of "Genetic Transformation and Green Fluorescent Protein Labeling in Ceratocystis paradoxa from Coconut"

_ijms, 2019, doi:10.3390/ijms20102387_

Round 1
Reviewer 1 Report
The Authors of ms ID: ijms-492857 identified a method for the genetic transformation of the fungus ascomycete Ceratocystis paradoxa, causal agent of black rot and stem-bleeding in the tropical fruit plants pineapple, banana, and coconuts. To confirm the stable integration of expression vector for GFP, the Authors successful performed PCR on samples of four different transformants. After subculturing several times, they conclusively detected green fluorescence of the four transformants and verified their pathogenicity on wounded coconut fruits.
This reviewer recommends the Authors to improve the text by removing inaccuracies and typing errors, using the correction tool of word processor, and improve English language.
Comments and suggestions are in the following points.
Abstract
L15: insert “palm” after coconut, and Latin name “Cocos nucifera L.” after “coconut palm”
L22: delete “mainly”, insert “X-3314 strain” after “C. paradoxa”
Introduction
L67: insert commas after “acutatum” and “fungicola”
L68: insert a space after “macrospora”
Results
L77-78: do not use the acronyms for PDA and HmB for their first occurrence
L80: insert a space in “Figure1”
L79-82, L85-88 (caption of Figure 1), and L237: the different concentrations of HmB have to be referred by using the volume aliquots used in PDA plates if they are stock concentrations, because there is not agreement with L101-107, L282 and L286-287: clarify here if they are final or stock concentrations.
L92,99: did you count spores or protoplasts ?
L94,95: if possible, insert a Table with the performances of the six different protoplast release buffers
L98: check English
L120: insert concentration of HmB and spaces
L134: here and where they occur, correct typing errors, change “Conformation” to “Confirmation”
L148. delete the selection borders of CK image in Figure 6
L149: delete repetition of “Disease”
Discussion
L155: check “inform”
L168: correct typing errors here and where they occur, insert a space in “paradoxawas”
M&M
L197: insert “and plant material” in 4.1 subtitle
L198: move “(Collection No. CCTCC AF2014002)” after “strain”
L198-201: insert plant material: coconut fruits of “Malayan Red Dwarf”, as in L303
L207, 222: correct typing errors here and where they occur, insert a space in “selection[23].” and “μmfilter“
L221-223 and L223-225: incomplete phrases, then check brackets
L221, 257: check typing of “10 mL l.0 mol/L” and “6 mL of l×106/mL”
L257: change “was” to “were”
L225: insert a space in “buffer1”
L232: insert “PCR” before “primers”
L237, L79-82, and L85-88 (caption of Figure 1): the different concentrations of HmB have to be referred by using the volume aliquots used in PDA plates if they are stock concentrations, because there is not agreement with L101-107, L282 and L286-287: clarify if they are final or stock concentrations.
L237-243: correct typing errors by insertion of spaces
L245-250: correct typing errors by insertion of spaces
L250,260: Change “North America” and “US” to “U.S.A.”
L251: change font of “4.6. Protoplast preparation”
L263: check the number of release buffer, it was release buffer 2 in L90
L272: correct typing errors here and where they occur, insert a space in “150μL”, “600μL” etc
L272-280: correct typing errors by insertion of spaces
L314,315: check comma
L291: check English
L297,298: delete “reaction” because PCR is Polymerase Chain Reaction
Author Response
Dear Sir/Madam,
Following are our response about reviewer 1’ comment to our manuscript No ijms-492857
1 .According your kind advise, we corrected some typewriting errors which were appeared in our manuscript , inserted some spaces and commas after some words (L15, L22,L67-68,L77-78,L80,L120, L134, L148, L149, L168,L207, L221-223 and L223-225 ,et al.) and checked the English (L98).
2. We moved “(Collection No. CCTCC AF2014002)” after “strain”
3. In order to express better, we changed the sentence of L155 high lighted on the manuscript.
4. We inserted “and plant material” in 4.1 subtitle, inserted plant material: coconut fruits of “Malayan Red Dwarf”, as in L303
5 .Yes,we count spores or protoplasts by hemocytometer Counting.( L92,99: did you count spores or protoplasts ?)
6 .We are sorry that we did not insert a table in L94,95 with the performances of the six different protoplast release buffers .Because we are preparing another manuscript about “Research on the Preparation and Regeneration Condition of the Ceratocystis paradoxa Protoplast”,which refer to the table. If used in this paper(No.ijms-492857) ,we think it will be incomplete. So after deep consideration, we did not insert the table. Thanks for your understanding.
7. In this paper, the stock concentration of HmB is 1g/20mL, but according to the request of experiment, we adjust to different concentrations of HmB, so they are final concentration in the manuscript.
8 .We apologized for our poor English. We will try to improve it in revision.
All the revisions were high lighted on our manuscript.
Thank you very much.
Best regards
Yours sincerely
Xiaoqing Niu
Reviewer 2 Report
This paper describes development of a transformation system for an economically important pathogen. Several points need to be clarified including the name of the pathogen, prior work, and presentation of results. Given the 2014 Mycologia paper (Mbenoun et al) that revised the larger Ceratocystis paradoxa group, you must verify which species you are working with. The current name could be Ceratocystis ethacetica, but that can be determined with some minimal sequencing. This information could also change the host range information presented in lines 53-56.
A quick perusal of the internet brought up a 2015 thesis report (University of Malaysia Sarawak) of transformation of Ceratocystis paradoxa using Agrobacterium-mediated transformation. That report also does not specify which species was used in the new revised nomenclature. You may want to mention this report.
Figure 1 legend is high redundant. Also need to write out Ceratocystis in first sentence. Suggested wording for second sentence "The growth of C. paradoxa on PDA medium with 10 ug/ml, A; 20 ug/ml, B; .....
Figure 3 results/legend are confusing. CK=wild type and wild type was cultured on 40 ug/ml HmB. However all three plates of CK appear to have a full plate of mycelial growth, even though the legend and text (line 108) state that the wild type failed to grow. Need to change the photos or the legend.
Figure 5 is not necessary. The information is mentioned in the text.
Figure 6 CK needs cleaning. It still contains the selection points.
The primer sequence, lines 232-235, should be moved to the beginning of section 4.9.
Reference 1, Alfieri, S.A. is incomplete. Need to include Florida Dept. of Agriculture.
Suggested rewording of abstract "...As. the mechanism of pathogencity of C. paradoxa has not been determined, an exogenous gene marker was introduced into the fungus.In this study, pCT74-sGFP which contains green fluorescent protein (GFP) gene, and the hygromycin B resistance gene as a selective marker, was used as an expression vector. Several protoplast release buffers were compared to optimize protoplast preparation. The plasmid pCT74-sGFP was successfully transformed into the genome of C. paradoxa, which was verified using polymerase chain reaction...."
Author Response
Dear Sir/Madam,
Following are our response about reviewer 2’ comment to our manuscript No ijms-492857.
1 We identified this pathogen Ceratocystis paradoxa causing coconut bleeding disease is in 2012. Mainly by morphological observation and molecular method molecular biological techniques. The host rage mentioned in this manuscript contained which was mentioned in 2014 Mycologia paper.
2 And we mentioned the 2015 thesis report in our manuscript, and some reference orders changed marked in our paper.
3 We shorten Figure 1 legend fllowed your advice.
4 To mention it, CK in Figure 3 is empty (the fungus did not grow), maybe the light reason when we took photo, it appear to have a full plate of mycelial growth. In order to clarify the fact, we repeat the experiment and changed a new photos.
5 we deleted the Figure 5 followed your guide
6 Figure 6 CK was cleaned.
7 We moved to The primer sequence to beginning of section 4.9.
8 we complete the reference 1
9 we reword of abstract. Thank you very much.
Best regard
Yours truly
Xiaoqing Niu
Reviewer 3 Report
The authors describe a transformation method for Ceratocystis paradoxa, a major pathogen of coconut. This method will help researches better understand the pathogenicity of this fungus and might also act as guidance for transformation of related fungi.
The manuscript is overall well written and I think only a few concerns need to be addressed before publicaton which I have listed below, the most important one is Line 144/145.
Line 43: the comma between producers, researchers should be an and
Page 3 captation of Figure 1: the text could be shortened, as all plates are PDA medium with the fungus therefore A-D should only mention which antibiotic concentration was used
Line 107/110: the antibiotic "concentrations" are given as how much of a certain was concentration was added to a specific volume of media, this should be calculated into how high was the final concentration of the antibiotic in the media as a final concentration. Also this should be consistent thrughout the whole document
Line 144/145: the manuscript would benefit if the data would be included as a Figure
Figure 6: The bottom panel has the manipulation tools from power point or similar visible this should be changed for publication
Line 185: There is no data presented that these transformants perform any better than the ones mentioned in the literature (see Line 144/145), therefore it is even mor important that GFP images are shown of fungus infected plants.
Line 251: the font type of the heading is different from the one use for the manuscript
Author Response
Dear Sir/Madam,
Following are our response about reviewer 3’ comment to our manuscript No ijms-492857.
1 we changed comma between producers, researchers as and
2 About the concentrations, in this whole document, the concentrations of HmB is final concentrations,and its stock concentration is 1g/20ml . In figure1, And every plate was 10mL PDA with 2μL, 4μL,6μL,8μL of HmB(1g/20mL),means the final concentration of HmB was 10μg/mL,20μg/mL, 30μg/mL,40μg/mL. And in 2.3 Screening and detection of transformants, we made a big mistakes because of our carelessness.
3 We supplement as a Figure about Line 144-145
4 we changed the font type in Line 251
Thank you very much.
Best regards
Yours sincerely
Xiaoqing NIU